# Using Particle Residence Time Distributions as an Experimental Approach for Evaluating the Performance of Different Designs for a Pilot-Scale Spray Dryer

**Zelin Zhou** * , **Timothy A. G. Langrish** and **Sining Cai**

Drying and Process Technology Group, School of Chemical and Biomolecular Engineering, Building J01, The University of Sydney, Darlington, NSW 2006, Australia
* Correspondence: zzho3224@uni.sydney.edu.au

**Abstract:** The performances of four different designs for a pilot-scale spray dryer have been evaluated and compared based on experimentally measured particle residence time distributions (RTD), recovery rates and physical properties of spray-dried fresh skim milk. The RTDs have been measured using a dye pulse injection method, and the measurements have been fitted to models using continuous stirred-tank reactors in series (CSTR-TIS) for quantitative performance evaluation and comparison. Conical drying chambers and a box connection design have been used in the latest dryer design to reduce the amount of wall deposition and provide a smoother gas flow pattern. The particle-to-gas mean residence time ratio for the latest design is significantly closer to unity (1.6 s/s to 1.0 s/s) compared with earlier designs (2.6 s/s to 1.5 s/s). The latest design has a wider spread of RTD ($n = 5$–8) compared with earlier designs ($n = 13$–18), which may be linked to the recirculation zone in the box connection. Although the latest design has a wider spread of RTD, the conical design has shown promising results compared with a cylindrical drying chamber in terms of overall wall deposition behaviours.

**Keywords:** spray drying; wall-deposit re-entrainment; particle residence time; chamber design

## 1. Introduction

Spray dryers are drying devices used in many industries, including food manufacturing, chemical production, and pharmaceutical applications. Spray drying has been used in the drying process for many thermally sensitive products [1,2]. For thermally sensitive products, the quality of the final products is significantly affected by their integrated thermal time exposure during thermal processing [3]. In the case of spray drying, the extent of thermal exposure (temperature) is mainly dependent on the operating conditions, including (but not limited to) inlet gas temperatures and feed-to-drying gas ratios. The impact of operating conditions in the spray-drying process on the quality of the final product is relatively well studied [4–6]. The duration of the thermal exposure, also known as the particle residence time, also depends on both the characteristics of the spray dryer and the properties of the product itself. It may be desirable to optimise and redesign parts of the spray dryer to improve the performance of the spray dryer in terms of the quality of the final production and energy consumption.

Reducing wall deposition is one of the main objectives of a possible redesign process for spray dryers [7]. Wall deposits are particles that deposit on the inner walls of spray dryers. Wall deposits of particles not only contribute to the loss of products from the solids outlets of the equipment but also tend to react or change and then re-entrain in the gas flow, which in turn affects the quality of the spray-dried products. The wall deposition and re-entrainment process may significantly extend the particle residence time in the spray dryer [4,8,9]. The extended particle residence time leads to greater thermal degradation in the final product, especially for materials that are more sensitive to heat.

Traditionally, the optimisation process is mainly based on experimental trials or empirical knowledge, which may be either time-consuming, expensive and/or less reliable [10,11]. With advances in computational fluid dynamics (CFD) simulations, CFD has been widely used to study the gas flow pattern within spray dryers and has assisted in redesigning parts of the equipment [7,12,13]. CFD simulations have provided significant insight into the gas flow patterns within spray dryers. However, CFD simulations require careful validation due to uncertainties in turbulence modelling, and they also have significant computational-time requirements. Furthermore, particle-wall interactions, particularly the wall deposition and re-entrainment process, are sometimes highly simplified in CFD studies of the spray-drying process [14,15], although more sophisticated approaches have been suggested [16].

As mentioned previously, the particle residence time is affected by the particle-wall interaction. Experimental validations of the results from CFD simulations are essential when redesigning spray dryers. Jeantet et al. have suggested that particle residence time measurements are powerful tools that can provide an experimental overview of the performance of a spray dryer design under different operating conditions [3]. An experimentally measured particle residence time reflects the combined effect of the chamber design and the physical properties of the materials being spray dried under different operating conditions. By combining the CFD-assisted design process and the validation from experimentally measured particle residence times, the optimisation and redesign process of the spray dryer may be more efficient and reliable.

A few studies have experimentally measured the particle residence times in spray dryers [3,10,17–20]. In general, the particle residence times are significantly longer than the gas residence time, most likely caused by particle-wall interactions [9]. However, in the studies mentioned above, only one drying chamber design has been used for each study. Thus, the effect of different chamber designs on the measured particle residence time distribution has not yet been investigated. This study aims to improve the drying chamber designs further to achieve a lower extent of wall-particle interaction and improve the final product quality. The performance of different spray dryer designs has been evaluated based on the measured particle residence time distribution and the properties of the final spray-dried products.

## 2. Materials and Methods

### 2.1. Chemicals

Fresh skim milk (99% fat-free) was purchased from a local supermarket (Coles, Australia) in 1 L bottles. All fresh skim milk was stored at 4 °C for no longer than three days. Red food dye (Pillar box red, Queen, Australia) was also purchased from the same supermarket.

### 2.2. Spray Drying Processes

The fresh skim milk used for the experiments was brought back to room temperature (20 °C) prior to the experiments. A Buchi two-fluid atomiser with a nozzle diameter of 0.2 mm was used for all the experiments. The feed flow rate was 13.5 mL/min, and the atomising air flow rate was 10,000 mL/min at 2.2 bar. Three fans were operated in a push-pull configuration providing a drying gas flow rate that varied from 98 $m^3$/h to 315 $m^3$/h, depending on the fan speed as set by a variable speed drive. At least two repetitions were carried out for each combination of fan setting and configuration of the spray dryer.

### 2.3. Designs of the Spray Dryer

In a previous study by Langrish et al. [7], CFD was used to assist the redesign process of the drying chamber for a pilot-scale spray dryer in reducing the amount of wall deposition within the spray dryer. As a result, four different drying chamber designs have been developed and built (Figures 1–3). In the first two designs, two drying chambers

are connected via either four-inch or six-inch circular pipe connections. In the third design, drying chambers are connected by a box structure, as a CFD simulation suggested that the box design may lead to lower wall deposition fluxes compared with the first two designs [7].

Preliminary investigations have shown that a significant amount of wall deposition occurs in the first two drying chambers of the spray dryer [7,21]. This phenomenon is likely caused by the direct contact between the spray and the inner wall of the spray dryer. Thus, in the fourth design, the first two drying chambers have been redesigned as a conical shape to fit better the conical spray pattern produced by the two-fluid atomiser. The expansion angle of the conical section has been chosen to be 8.5°, which was designed to avoid flow separation due to the expansion while keeping the footprint of the spray dryer within reasonable limits. The box connection between the two drying chambers has also been redesigned accordingly. Except for the latest design (Design 4), drying chambers 1, 2, 3, 6, 7 are shared between all the configurations. Key differences between the designs and their internal volumes are shown in Table 1.

**Table 1.** Key differences between the designs and the internal volumes of the different designs.

| | Four/Six-Inch Cylindrical Connection | Box Connection | Conical Drying Chamber |
|---|---|---|---|
| Drying column 1 |  |  |  |
| Drying column 2 |  |  | |
| Type of connection between drying columns |  Cylindrical |  Box | |
| Internal volume (m$^3$) | 0.228/0.230 | 0.262 | 0.765 |

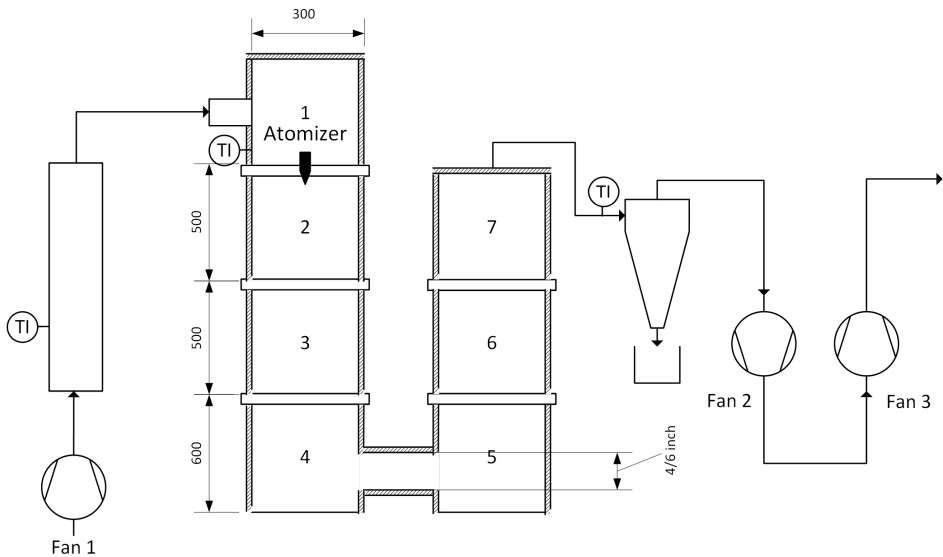

**Figure 1.** Spray dryer with cylindrical connections (Design 1: four-inch connection, Design 2: six-inch connection). All unit are in mm unless stated otherwise.

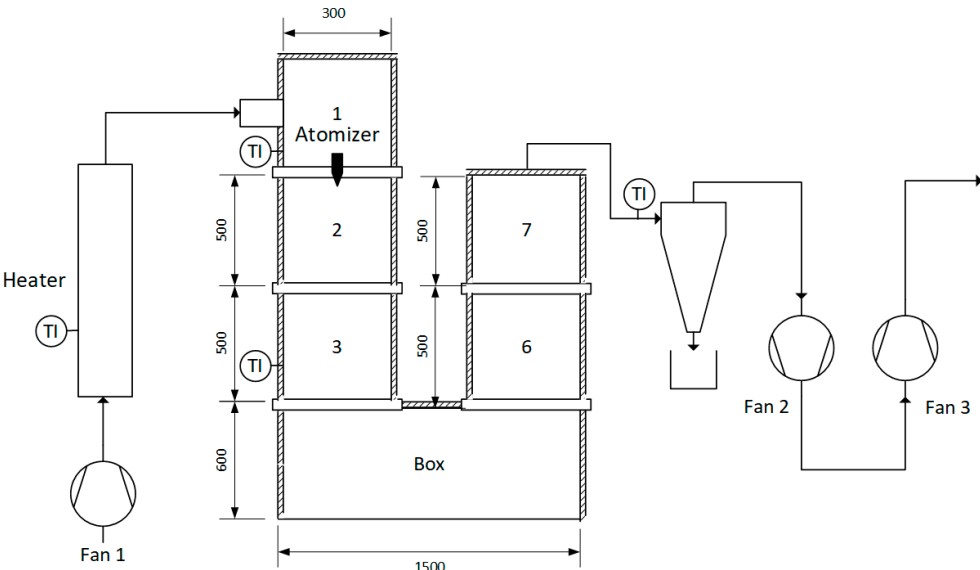

**Figure 2.** Spray dryer with cylindrical drying chamber and box connection (Design 3) [22].

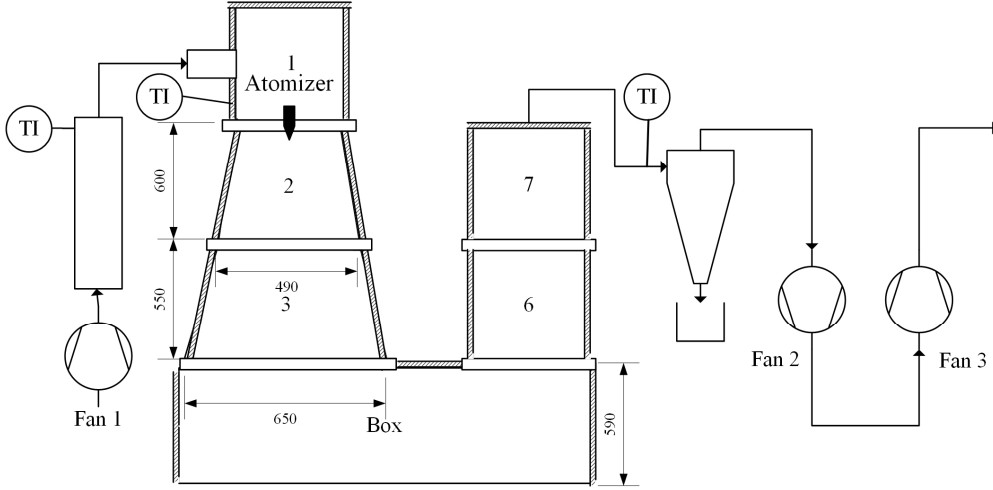

**Figure 3.** Spray dryer with conical drying chamber and box connection (Design 4).

### 2.4. Particle Residence Time Measurement

2.4.1. Particle Residence Time Measurement Device

The particle residence time measurement is the same device described in the study of Zhou and Langrish [22] (Figure 4). The particle residence time measurement device consisted of two parts. The first part was the automated dye injection system. The automated dye injection system consisted of a small liquid pump (5 mm × 4 mm × 4 mm), a valve, and a single-board computer to control the whole injection system. The automated dye injection system injected dye into the feed every 180 s during the experiments. Changes in tracer concentration were measured as the changes in the intensity of the detected laser light as the particles, with and without tracer, scattered the laser differently. The intensity of the scattered light was measured using a photomultiplier tube (Electron Tubes Limited, London, UK) at a sampling frequency of 10 Hz.

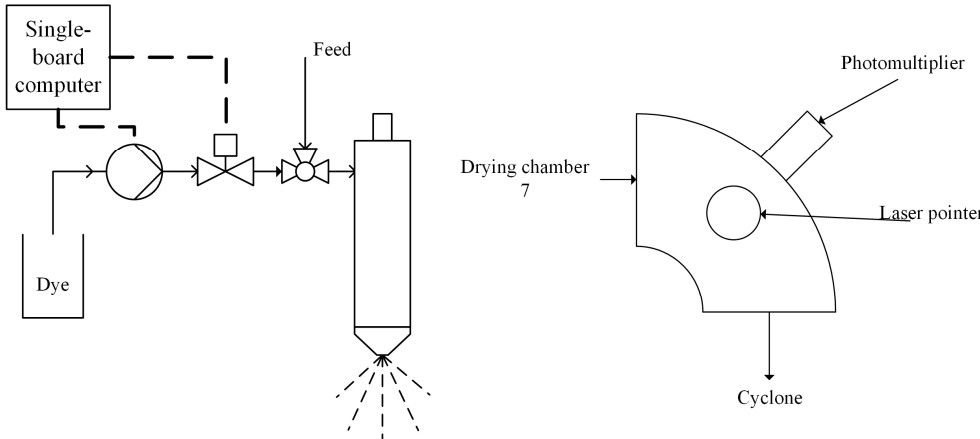

**Figure 4.** Schematic diagram of the RTD measurement devices [22].

The effect of different dye quantities has been investigated using manual injection prior to switching to an auto-injection system. Injection volumes ranging from 1 mL, 1.5 mL, and 2 mL (over the same period of time) have been tested using Design 3 at a gas flow rate of 257 m$^3$/h. We found that different amounts of dye injection mainly affect the signal-to-noise ratio. With smaller injection volumes, the signals measured were difficult to separate from the baseline (i.e., poor signal-to-noise ratio). In terms of particle residence time distribution, the fitted parameters for experiments with different injection volumes using manual injection are similar to those measured with auto-injection systems (Table 2). The mean particle residence times for experiments with manual injection were slightly longer than those with automatic injection systems. This difference is likely caused by human reflection time and the difference in tubing for dye delivery (longer tubing was used in manual injection experiments for safety reasons). It is reasonable to suggest that the quantity of dye injected into the chamber has a limited impact on the particle RTD measured, considering the variations caused by manual injections.

**Table 2.** CSTR fitting results for RTD measurement using different injection volume and methods.

|  | $\tau$ (s) | $n$ | Injection Volume (mL) |
| --- | --- | --- | --- |
| Manual injection | 11.9 | 15 | 1 |
|  | 11.4 | 10 | 1.5 |
|  | 11.5 | 15 | 2 |
| Auto injection | 9.3 ± 1 | 13 ± 4 | 2 |

2.4.2. Fitting the Particle Residence Time Distributions

To quantitatively analyse the measured signals from different configurations and operating conditions, all signals were normalised and fitted with a continuous stirred-tank reactor tank-in-series (CSTR-TIS) model (Equation (1)). The model parameters were fitted using MATLAB 2022a based on the least-squares method.

$$RTD(t) = \frac{n}{(n-1)!} \times \exp\left(-n \times \frac{t}{\tau}\right) \times \left(\frac{n \times t}{\tau}\right)^{n-1} \tag{1}$$

where $\tau$ is the overall mean particle residence time, and $n$ is the number of equivalent reactors in series [23].

2.4.3. Analysis of the Fluctuations in Signal

Normalised signals were transformed from the time domain to the frequency domain using Fourier transform to analyse the frequency of fluctuations observed in the measured signals. The last 60 s of each measured signal were analysed to reduce the interference from the main peak. Fourier transformations of the signals were performed using the fast Fourier Transform (FFT) algorithm available in MATLAB 2022a.

*2.5. Particle Size Distribution*

The particle size distribution may be affected by the operating conditions of the dryer, the atomiser, and the design of the spray dryer. Thus, the particle size distribution was determined for samples collected from different operating conditions. The particle size distribution was measured using a Malvern Mastersizer 3000 (Malvern Instruments, Malvern, UK) with a universal feeding funnel and a 100% feed flow rate at 2.2 bar.

*2.6. Scanning Electron Microscopy (SEM)*

The particle morphology was observed using Scanning Electron Microscopy (SEM). Samples were fixed on the sample stem using carbon tape. Then, the sample was coated with gold using a Quorum-SC7620 Mini Sputter Coater (Quorum Technologies, Lewes, UK). Images of the coated samples were observed at $1000\times$, $3000\times$ and $5000\times$ magnification with a Phenom-Prox SEM (Phenom-World, Eindhoven, The Netherlands).

*2.7. Moisture Content Measurement*

The moisture content of the samples was measured using the loss-on-drying method described in Ozmen and Langrish [24]. In summary, approximately 1 g of the collected samples was placed on a Petri dish and placed in a drying oven at 85 °C for a period of 23 h. The moisture content of the samples was calculated based on the change in mass.

**3. Results and Discussion**

*3.1. Mean Particle Residence Time—τ*

As shown in Figure 5a, the mean particle residence increases as the gas flow rate decreases. This observation is related to the fact that the particle residence time is mainly affected by the gas residence time and the gas flow rate. During the spray drying process, high-velocity droplets from the atomiser rapidly approach the drying gas velocity [25]. Thus, the particles are expected to have a similar residence time to the gas if there are no particle-wall interactions, and the gas residence time would normally be expected to be inversely proportional to the gas flow rate. Due to the differences in the designs, mainly their different dimensions (Table 1), the mean gas residence time for each design is different at any given gas flow rate. The difference in the internal volumes of Designs 1 and 2 is negligible compared with the differences between the other designs. Therefore, similar gas and particle residence times should be expected for the first two designs, which are consistent with the observed values. The internal volume of Designs 3 and 4 are significantly

larger than the first two designs, and thus longer mean gas and particle residence times are expected.

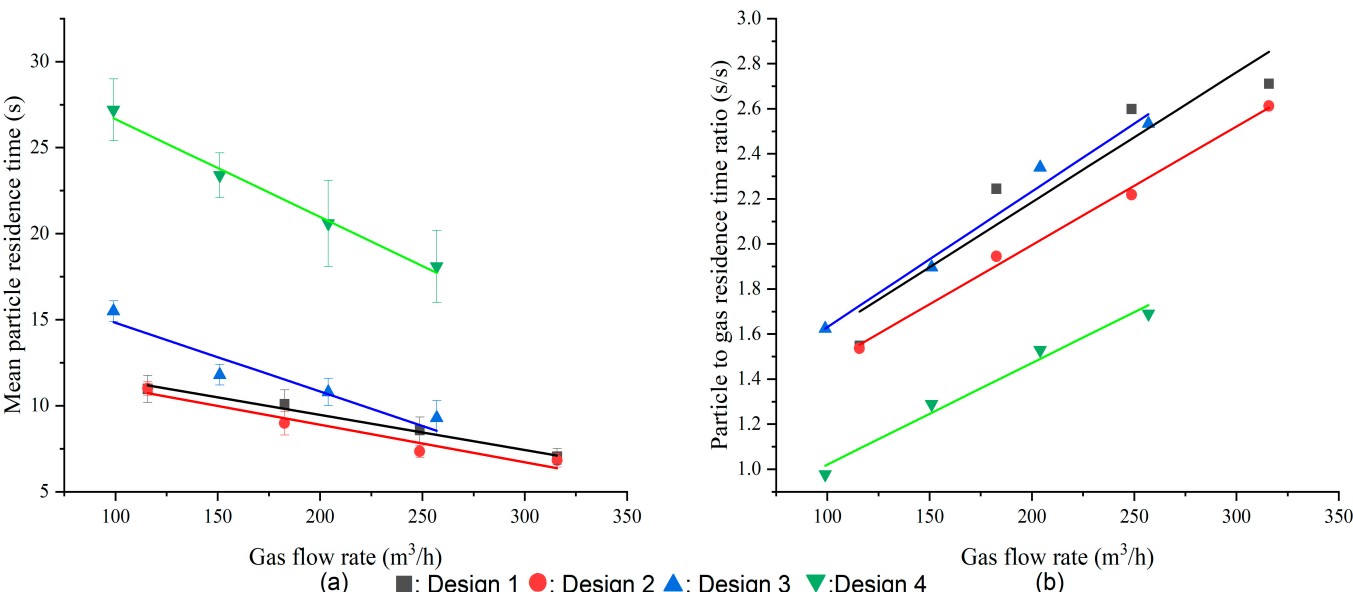

**Figure 5.** Mean particle residence time ($\tau$) and ratios between particle and gas residence time for different configurations and gas flow rates (**a**) mean particle residence time (**b**) particle to gas residence time ratio.

As discussed previously, the particle residence time is mainly affected by the gas residence time; it is undesirable to evaluate the performance of different designs solely on the basis of the absolute particle residence time. In practice, the particle residence time is expected to be significantly longer than the gas residence time due to particle-wall interactions. In addition to the operating conditions and the properties of the feed, the design of the spray dryer also plays an important role in determining the particle residence time. In this study, the feed and inlet gas temperatures are the same for all the designs. The difference between the mean particle and gas residence time ratios mainly depends on the dryer designs. Therefore, the ratio between the particle residence time and the estimated residence time of the gas has been used here to evaluate the performance of different designs. In other studies, the particle-to-gas residence time ratio is also called the chamber coefficient ($\varepsilon$) and was found to be related to the design of the spray dryer [26]. The gas residence times for different designs were estimated based on the gas flow rate and the internal volume of each design.

As shown in Figure 5b, Design 1 has the highest particle-to-gas residence time ratio of all the designs for most gas flow rates. This observation is related to the suboptimal connection design between the two drying columns. The four-inch connection between the two drying columns created a particle recirculation zone at the entrance and the exit of the connection due to sudden contraction and expansion [7]. The increased diameter (6 inches) of the connection in Design 2 has somewhat reduced the particle recirculation behaviour, and a small decrease in the particle-to-gas residence time ratio was observed. Unifying two drying chambers via a box connection (Design 3) aimed to avoid sudden changes in gas velocity due to sudden contraction and expansion. However, the observed mean particle-to-gas residence time ratio for Design 3 is slightly higher than the first two designs at the same gas flow rate. This result is likely to be related to the recirculation zone observed at the bottom left of the box in previous CFD simulations by Langrish et al. [7]. The recirculation zone may extend the mean particle residence time and widen the spread of the particle residence time distribution. In addition to the bottom-left corner, the particles are likely to hit the top plate of the box near the bottom of the second drying column and the right inner wall. Similar behaviour has also been observed in other

CFD simulation studies [27]. Despite having a slightly higher particle-to-gas residence time ratio, the box design is still promising due to its smoother gas flow pattern [7].

In a previous study by Langrish et al. [7], most wall depositions were found within the first two drying chambers, except the bottom of the box chamber. This phenomenon is likely to be due to spray from the atomiser hitting the inner wall of the dryer directly, which is caused by the mismatch between the spray pattern (conical) and the geometry of the drying chamber (cylindrical) (Figure 6). Changing the shape of drying chambers 2 and 3 from cylindrical to conical reduces particle-wall interactions, since a conical shape can better fit the spray pattern produced by the atomiser. This expectation is supported by the observed particle-to-gas residence time ratio for Design 4 being significantly lower than the previous designs at all gas flow rates. It is also worth mentioning that at the lowest gas flow rate, the average particle-to-gas-residence time ratio for Design 4 is slightly less than unity. This observation is likely due to the high initial velocity of the droplets coming out of the atomiser. Similar behaviour has also been reported in other studies [13,20].

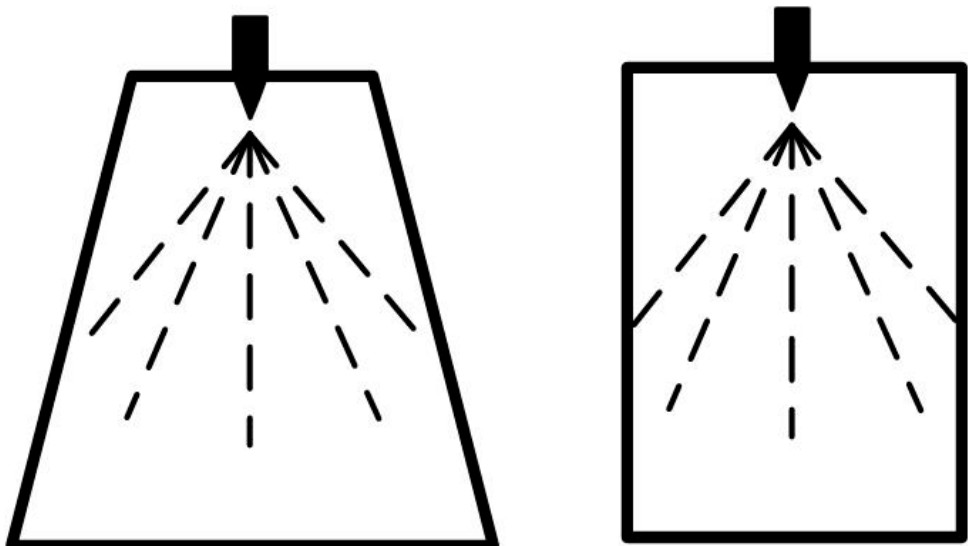

**Figure 6.** Difference between cylindrical and conical drying chambers.

### 3.2. Spread of Particle Residence Time—n

Other than the average length of time for the particles to stay in the dryer, the spread of the particle residence time is another important performance indicator. A broader spread in particle residence time distribution means a larger portion of the spray-dried products will stay longer than the mean particle residence time. A portion of spray-dried products having a longer residence time than others can affect the quality of the final product adversely. Therefore, the tighter the distribution, the better the performance. The spread of the particle residence (RTD) is described by the number of equivalent continuous stirred-tank reactors (CSTR) in series ($n$). A higher value for $n$ corresponds with a tighter spread in the residence time (i.e., closer to the behaviour of a plug-flow reactor, PFR) and vice versa. As shown in Figure 7, different gas flow rates have limited impact on the spread of particle residence time for a given chamber design ($p > 0.05$). The difference in the spread of particle residence times for the first three designs is also small ($p > 0.05$). This observation is likely to be due to the similarities in their designs (i.e., chambers 1, 2, 3, 6 and 7 are shared for the first three designs). The spread of the RTD in Design 4 is significantly greater than in the first three designs ($p < 0.05$). The broader spread of the RTD in Design 4 may be partly due to the recirculation zone near the bottom left of the box connection mentioned in previous studies [7]. The recirculation zone at the bottom-left section of the box connection will be the next point of development for improved designs. The conical chamber design may also cause extra re-circulation and widen the spread of the RTD in Design 4. However, this hypothesis does require confirmation from CFD simulations for Design 4, and it will be

the focus of future studies. Last but not least, the spread of the RTD for Design 4 showed a promising trend (i.e., increasing *n*) as the gas flow rate increased despite having the widest spread.

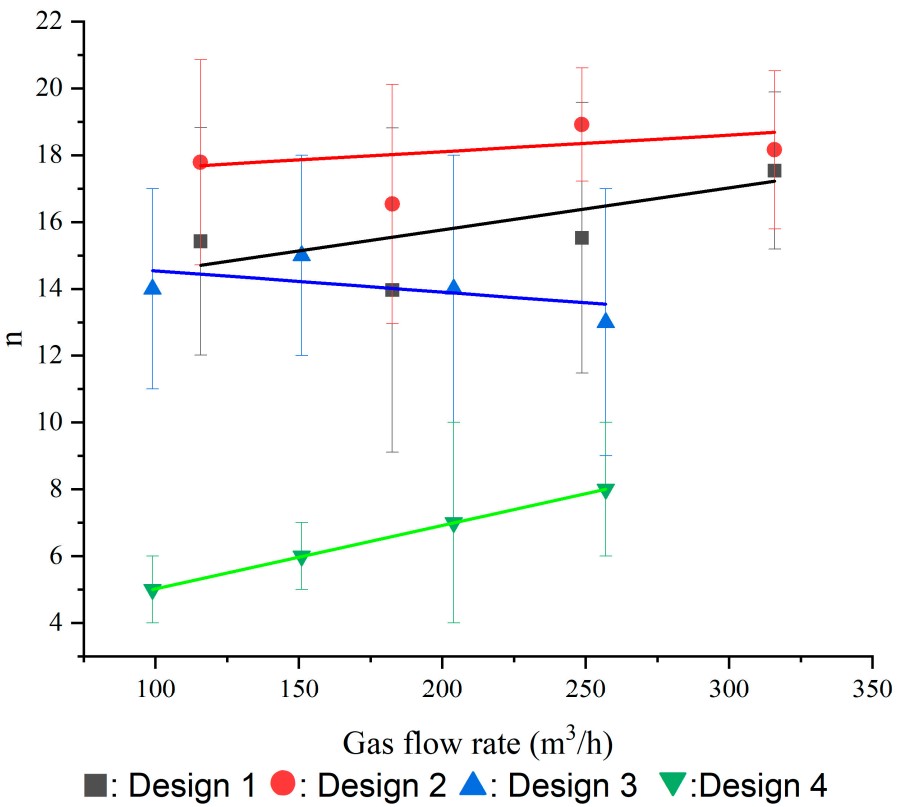

**Figure 7.** Number of equivalent CSTR in series (*n*) for different configurations and gas flow rates.

### 3.3. Fluctuations in Measured Signals

As shown in Figure 8, negative values at the initial part of the measurement were observed. They are likely caused by the short disruption in the feed to the atomiser caused by the introduction of the red dye solution, and this has a limited impact on the RTD measurement. Fluctuations were also observed in all measured particle residence time distribution (RTD) signals. The noise from the photomultiplier itself was sampled at an early stage of the experimental design. The relative magnitude of the noise from the photomultiplier is approximately 1% of the baseline values, while the fluctuations observed are around 5% of the baseline values. Furthermore, fluctuations in measured signals were also observed in other studies that used scattered light methods [23]. As a result, it is not likely that the fluctuations are inherent to the measurement system, and it is more likely to be linked to physical processes occurring within the dryer, including flow-stability issues or wall-deposition-related issues. Wall deposition and their re-entrainment process is a well-known physical process that occurs during spray drying processes [8,9]. Wall deposits that are re-entrained may be different in size and shape due to aggregation [28]. In addition to the shape and size, re-entrained wall deposits may also contain tracers. As a result, wall depositions refracted the laser differently and caused fluctuations in the measured RTD signal. Fourier transforms for the last 60 s of all the measured signals were performed using fast Fourier Transform (FFT) using MATLAB 2022a. Since the sampling duration is 60 s, any signals with a frequency lower than 1/60 Hz cannot be properly measured. According to the Nyquist-Shannon sampling theorem, to accurately characterize any waveform from a time series of data, the sampling frequency must be twice the signal frequency [29]. Therefore, any signals with a frequency higher than 5 Hz were ignored, since the sampling frequency is 10 Hz. The frequency with the highest amplitude in each measurement has been defined as the dominant frequency in the signals. The averaged

values of the dominant peaks for all combinations of gas flow rates and configurations are shown in Figure 9.

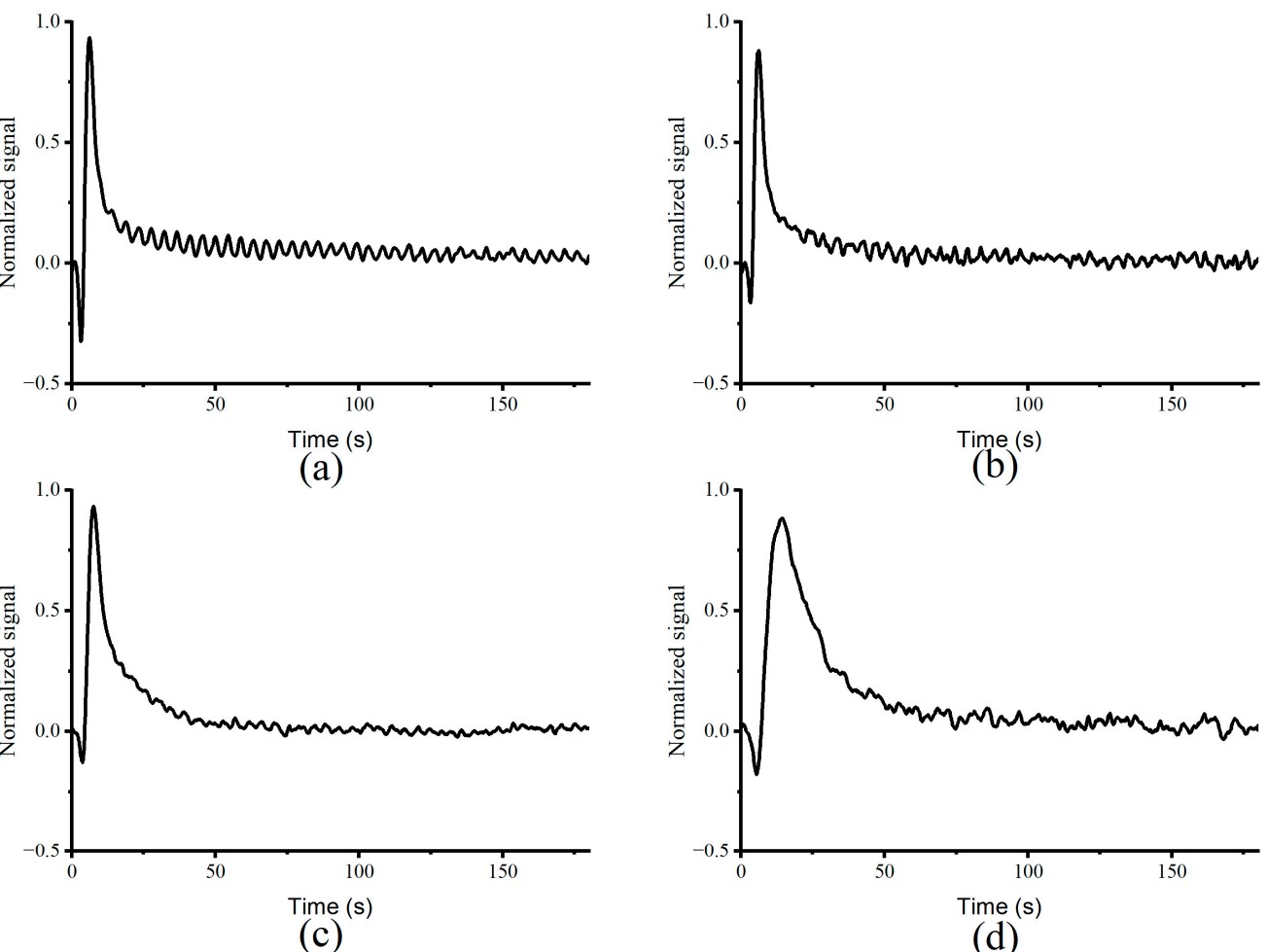

**Figure 8.** Averaged normalised signals for all designs at maximum gas flow rate (257–310) m$^3$/h; (**a**) Design 1: Four-inch connection; (**b**) Design 2: Six-inch connection; (**c**) Design 3: Box connection; (**d**) Design 4: Conical drying chamber.

### 3.3.1. Flow Related Factors

It is unclear whether the fluctuation is caused by flow-stability-related issues. If the flow is time-dependent or transient, the fluctuations observed in signals may be related to the oscillation in the flow. A dimensionless parameter can describe the oscillation behaviour of the flow, called the Strouhal number (*Sr*), which can be expressed as a function of the Reynolds number [30–32]. Reynolds numbers in different parts of the spray dryer in this study are within a range ($10^3 < Re < 10^5$) where the Strouhal number is approximately constant ($Sr \approx 0.2$) [30,31]. The *Sr* calculated based on the frequency measured is within a range of 0.04 to 0.71, which is different from the *Sr* for flow oscillation estimated based on the Reynolds number of around 0.2. Furthermore, when the Strouhal number is constant, the oscillation frequency of the vortex shedding is proportional to the gas velocity for a given characteristic length. In this study, the frequency of the fluctuations is independent of the gas flow rate and thus the gas velocity for the same configuration (Figure 9). These observations suggested that the fluctuations in the signals are not simply related to any oscillations in the flow patterns.

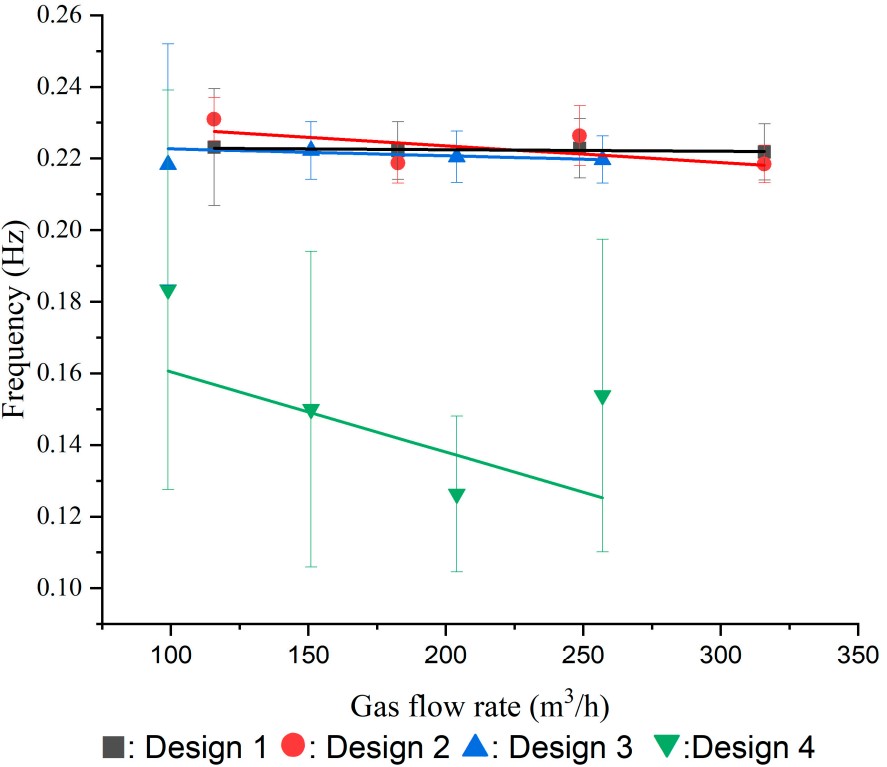

**Figure 9.** Dominant frequencies for different configurations and gas flow rates.

3.3.2. Wall Deposition and Wall-Deposit Re-Entrainment-Related Factors

Another plausible explanation for the fluctuations observed is that the fluctuations are related to wall deposition and re-entrainment processes. There is no significant difference between the dominant frequencies in the first three designs ($p > 0.05$). This observation may be related to the fact that drying chambers 1, 2, 3, 6 and 7 are shared between these three designs. The dominant frequencies for Design 4 are slightly lower than those for the first three designs ($p < 0.25$). Like the fitted parameters (i.e., $\tau$ and $n$) for the CSTR-TIS model, larger variations in the dominant frequencies in Design 4 were observed, which may be related to the geometry of the drying chambers. This result suggests that the difference in fluctuation frequencies is likely linked to drying chambers 2 and 3, which are also the chambers where most wall deposition occurs. Factors affecting the wall deposition and the re-entrainment process are summarised and shown in Figure 10 [33].

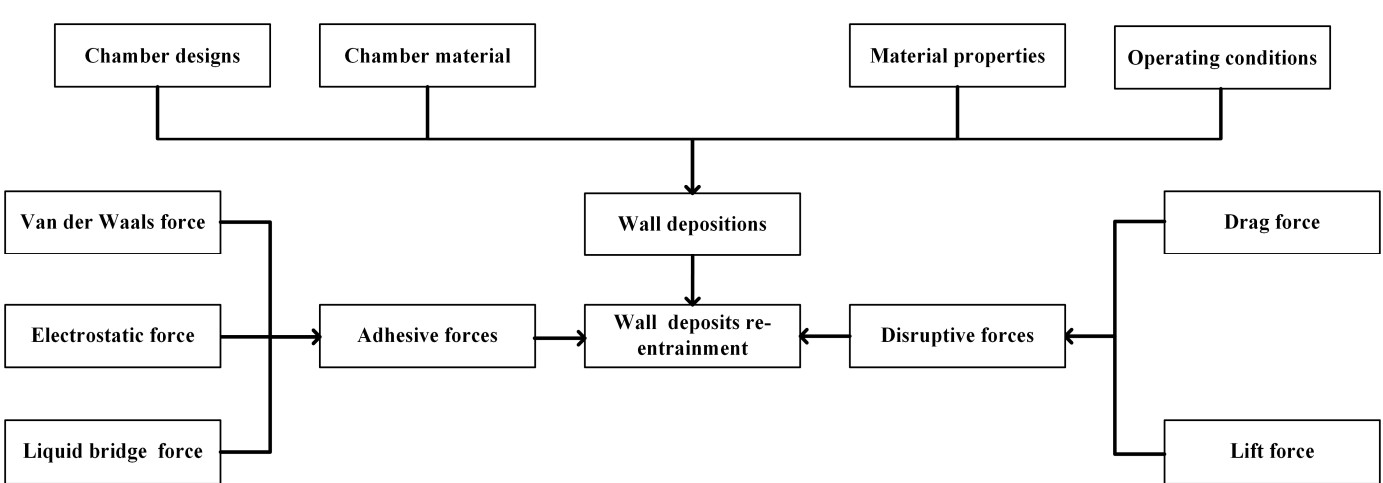

**Figure 10.** Factors affecting the wall deposition and wall deposit re-entrainment.

Other than the gravitational forces, one of the main disruptive forces (i.e., forces that remove wall deposits from the inner wall of the spray dryer) for the re-entrainment of wall deposits is the force from turbulent flow fluctuations inside the spray dryer. Within the boundary layer of turbulent flows, approximately 50% of the turbulence energy production occurs at the inner layer for the boundary layer in the form of turbulent bursts [34]. Therefore, it is important to estimate the turbulent burst frequency inside the spray dryer and compare it with the observed fluctuation frequencies. Here, in this study, turbulent burst frequencies have been estimated based on the data available in the literature since measuring the turbulent burst frequency is complicated and beyond the scope of this study. A few studies have proposed different methods for estimating the scaling behaviour of the turbulent burst frequency, including inner/outer variable scaling, mixed scaling, mixed modelling (inner and outer variables) and scaling based on the Taylor microscale [35–37]. Although the later methods may be better in estimating the mean turbulent burst frequency in the drying chamber, they were not implemented in this study because some of the variables required for implementing the latest estimation methods were not measured due to practical issues and are beyond the scope of this study. Last but not least, this study did not estimate the mean turbulent burst frequencies in the conical drying chambers due to their more complicated geometry. Additionally, it appears that no work in the literature has investigated the mean turbulent burst frequency in a similar geometry (i.e., a conical chamber). The turbulent burst frequency is estimated based on the inner-variables scaling method, which is the latest approach that is applicable to this study and shown in Table 3.

**Table 3.** Estimated turbulent burst frequency based on inner variables for Designs 1 and 2 (diameter = 0.3 m).

| Gas Flow Rate (m³/h) | *Re* | *f* (Hz) |
|---|---|---|
| 315 | 16,104 | 0.76 |
| 257 | 13,139 | 0.58 |
| 249 | 12,730 | 0.50 |
| 204 | 10,429 | 0.38 |
| 183 | 9356 | 0.29 |
| 152 | 7771 | 0.23 |
| 116 | 5930 | 0.13 |
| 99 | 5061 | 0.11 |

As shown in Table 3, the estimated turbulent burst frequencies are higher than those observed, except for those at the lowest gas flow rate. Considering that the wall deposition occurred during the operation of the dryer, the roughness of the inner wall surface may be higher than the values assumed here. Thus, the actual turbulent burst frequency is likely to be higher than the estimated values based on the inner variables. Although the estimated turbulent burst frequencies are greater than the frequencies observed, they are still within the same order of magnitude. The two frequencies are not identical because the re-entrainment process is likely to be hindered by the material being spray dried. Given this hypothesis about the turbulent bursts acting as driving forces for the re-entrainment process, it would be expected that the frequency of the bursts should be greater than the frequencies for the re-entrainment process because the material would be expected to dampen the frequency for the re-entrainment of the deposits. This hypothesised trend has indeed been observed here. In conclusion, it is reasonable to suggest that the fluctuations observed are most likely to be related to the wall deposition and re-entrainment process. The lowest wall-deposition re-entrainment frequency observed in Design 4 suggests that the conical chamber design has successfully reduced the chances of wall deposition occurring in the first two chambers.

### 3.4. Effects of Feed Properties on Re-Entrainment

To investigate the effect of different feeds on the particle residence time distribution and the observed frequencies, a salt solution (NaCl, 8.8 wt%) was used as the feed instead of fresh skim milk. Compared with skim milk, which generally gives amorphous lactose components in the spray-dried powders, salt gives crystalline powders, and it is less likely to deposit on the inner walls of the dryer [21,38]. The salt solution experiments were only performed for the two latter designs (Designs 3 and 4) at the maximum gas flow rate since they represent the effects of different drying chamber designs on the particle residence time distribution (Design 3 has cylindrical drying chambers, Design 4 has conical drying chambers).

As shown in Table 4, the mean particle residence time is shorter, when salt solution is used as feed, compared with fresh skim milk ($p < 0.05$). These results suggest that the mean particle residence time may be affected by the physical properties of the feed, which mainly affect the wall deposition and re-entrainment behaviour. Experiments using salt solutions as the feed also show a tighter spread than skim milk, but the difference may not be significant (Design 3: $p = 0.07$, Design 4: $p = 0.21$). The similar spread of the particle residence time distribution suggests that the spread is more dependent on the design of the drying chambers and recirculation and is less dependent on the material properties. The fluctuation frequencies observed when salt solution is used as the feed are slightly lower than those with skim milk (Design 3: $p < 0.01$, Design 4: $p < 0.3$). This observation further supports the hypothesis that the fluctuations in the signals are indeed linked to the wall deposition and re-entrainment process, since the fluctuations are different for the different feed materials.

**Table 4.** RTD measurement result summary for experiments with salt solution.

| Gas Flow Rate | Measurements | Design 3 | Design 4 |
|---|---|---|---|
| | Mean particle residence time ($\tau$, s) | 8.3 | 8.3 |
| | Mean gas time (s) | 3.7 | 3.7 |
| 257 m$^3$/h | Ratio between particle and air residence time (s/s) | 2.2 | 2.2 |
| | Number of equivalent CSTR in series ($n$) | 15 | 15 |
| | Fluctuation frequency (Hz) | 0.206 | 0.206 |
| | Re$_{min}$ | 15,945 | 15,945 |

### 3.5. Solid Recovery Rates and Physical Properties of the Spray-Dried Particles

The solid recovery rate is an important indicator of the performance of different designs of spray dryers because it is related to the throughput of the equipment. The recovery rate for Design 4 is slightly higher than other designs at most gas flow rates, but the difference may not be significant ($p > 0.05$). The solid recovery rates were observed to decrease with the gas flow rate observed for all the designs. This behaviour may be related to the design of the spray dryer. Unlike traditional tall-form spray dryers, the spray dryer used in this study has two vertical drying columns connected by a cylindrical or box connection to reduce the height of the dryer while providing sufficient residence time for the drying of the particles. Lower gas flow rates may not be sufficient for carrying larger particles from the bottom of the spray dryer to the outlet. This hypothesis is supported by the decreasing trend in mean particle diameters (Dx$_{50}$ and Dx$_{90}$) with the decrease in gas flow rates (Table 5). In addition to the small particle diameter, this hypothesis is also supported by observations in previous studies, where the highest wall deposition fluxes occur at the bottom of the drying chambers [7,15].

**Table 5.** Physical properties of spray dried powder collected from different designs and gas flow rates.

| Gas Flow Rate | Measurements | Four-Inch Cylindrical Connection | Six-Inch Cylindrical Connection | Box Connection | Conical Drying Chamber |
|---|---|---|---|---|---|
| 257–315 m$^3$/h | Moisture content | 3% | 2% | 3% | 3% |
| | Solids recovery rate | 43% | 51% | 48% | 51% |
| | D × 10 (μm) | 4.2 | 4.7 | 5.2 | 4.9 |
| | D × 50 (μm) | 8.4 | 8.9 | 11.8 | 10.0 |
| | D × 90 (μm) | 16.2 | 13.2 | 19.8 | 25.2 |
| 204–250 m$^3$/h | Moisture content | 3% | 2% | 4% | 3% |
| | Solids recovery rate | 33% | 37% | 41% | 41% |
| | D × 10 (μm) | 3.9 | 4.2 | 5.2 | 5.0 |
| | D × 50 (μm) | 7.0 | 7.3 | 11.5 | 9.6 |
| | D × 90 (μm) | 13.2 | 12.0 | 19.2 | 17.9 |
| 152–180 m$^3$/h | Moisture content | 3% | 3% | 5% | 4% |
| | Solids recovery rate | 14% | 13% | 17% | 23% |
| | D × 10 (μm) | 3.7 | 4.0 | 4.7 | 5.0 |
| | D × 50 (μm) | 6.6 | 6.4 | 9.1 | 9.2 |
| | D × 90 (μm) | 8.9 | 13.2 | 17.2 | 17.0 |
| 99–116 m$^3$/h | Moisture content | 6% | 4% | 7% | 7% |
| | Solids recovery rate | 7% | 11% | 8% | 6% |
| | D × 10 (μm) | 4.4 | 3.8 | 4.8 | 5.1 |
| | D × 50 (μm) | 6.3 | 6.2 | 8.3 | 8.0 |
| | D × 90 (μm) | 10.3 | 8.9 | 14.5 | 12.5 |

SEM images of the particles collected from different designs are shown in Figure 11. Shrinkage in the particles was observed for most of the particles collected, which is related to the low drying temperature (inlet gas temperature: 100 °C). The surfaces of particles dried at lower temperatures tend to stay moist for a longer period than the surfaces of the particles dried at higher temperatures. The moist surface of the particle allows shrinkage to occur during the spray drying process [39–42]. Other than the shrinkage in the observed particles, powders collected from Designs 1 and 2 have more small particles than those collected from Designs 3 and 4. This observation is also confirmed by the smaller particle diameter ($Dx_{10}$) for the particles collected from the first two designs. Since the same atomiser and atomising conditions were used across all designs and operating conditions, the difference in particle size distribution depends more on the design of the spray dryer rather than the atomiser. Smaller particles in the collected samples are probably caused by the collisions between the particles and the inner wall of the spray dryer. Smaller particle diameters were observed for the first two designs compared with the later designs, which is likely to be linked to the smaller connection between the two drying columns.

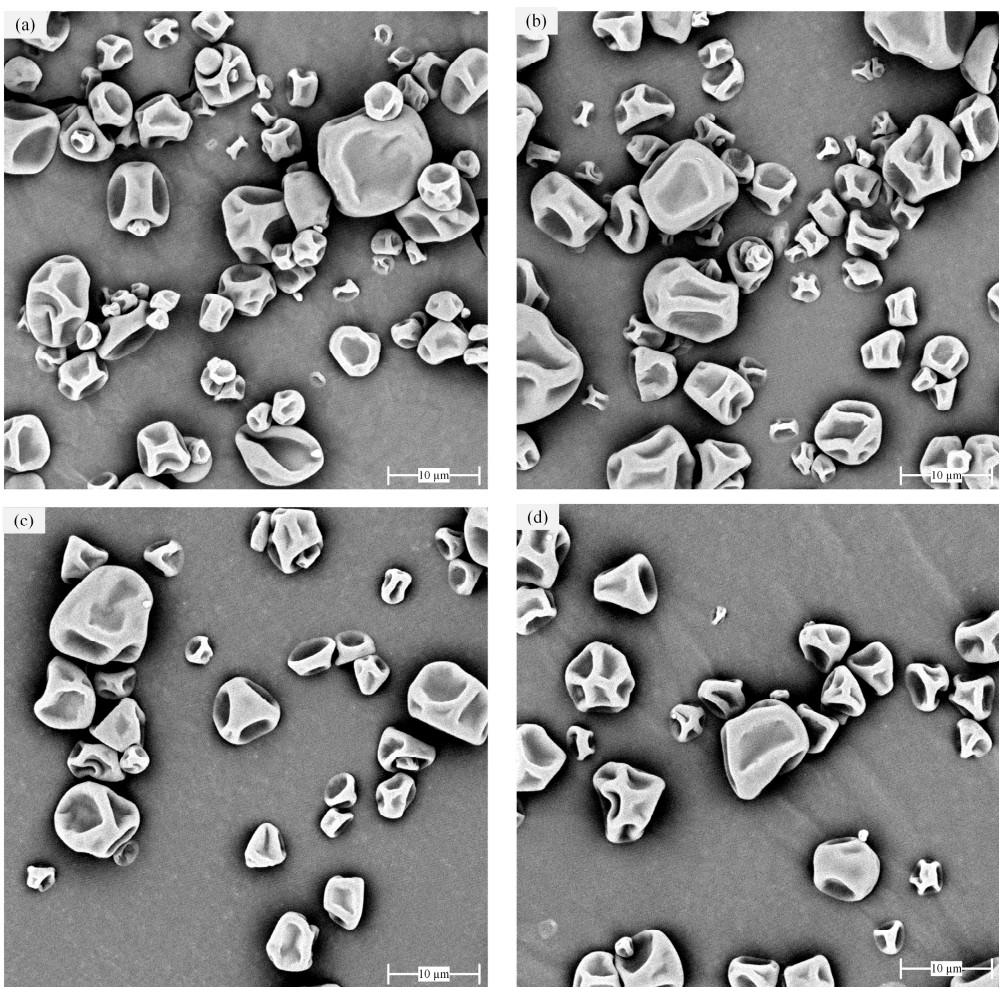

**Figure 11.** SEM image of particle collected from different designs with maximum gas flow rate; (**a**) Design 1: Four-inch connection; (**b**) Design 2: Six-inch connection; (**c**) Design 3: Box connection; (**d**) Design 4: Conical drying chamber.

## 4. Conclusions

In this study, the performances of four different designs of a pilot-scale spray dryer have been evaluated mainly based on the particle residence time distributions (RTD), solid recovery rate, and physical properties of the spray-dried products. The mean particle residence times increase as the gas flow rate decreases. The ratio between the particle and gas residence time varies significantly depending on the designs of the drying chambers and gas flow rates (Design 1: 1.5–2.7 s/s, Design 2: 1.5–2.6 s/s, Design 3: 1.5–2.5 s/s, and Design 4: 1.0–1.7 s/s). Later designs have a wider spread in the particle residence time distribution compared with the first two designs, and the spread is likely to be related to recirculation within the dryers. Fluctuations were observed in all measured signals, and they were probably linked to the wall deposition and re-entrainment processes. In conclusion, the results have suggested that the shape of the first two drying chambers plays an important role in both the particle residence time distribution and the wall deposit re-entrainment process. The conical chamber design appears to be superior to the cylindrical design. Optimising the design of the connecting chamber (i.e., the bottom chamber) to reduce recirculation could be a focus of future studies.

**Author Contributions:** Conceptualization, T.A.G.L. and Z.Z.; methodology, S.C., T.A.G.L. and Z.Z.; software, Z.Z.; validation, T.A.G.L. and Z.Z.; formal analysis, T.A.G.L. and Z.Z.; investigation, S.C., T.A.G.L. and Z.Z; resources, T.A.G.L.; data curation, Z.Z.; writing—original draft preparation, Z.Z.; writing—review and editing, Z.Z. and T.A.G.L.; visualization, Z.Z. and T.A.G.L.; supervision, T.A.G.L.; project administration, T.A.G.L.; funding acquisition, T.A.G.L. All authors have read and agreed to the published version of the manuscript.

**Funding:** This research received no external funding.

**Institutional Review Board Statement:** Not applicable.

**Informed Consent Statement:** Not applicable.

**Data Availability Statement:** The data that supports the findings of this study are available from the corresponding author upon reasonable request.

**Acknowledgments:** This research is supported by an Australian Government Research Training Program (RTP) Scholarship.

**Conflicts of Interest:** The authors declare no conflict of interest.

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
