# Peer review of "Using Particle Residence Time Distributions as an Experimental Approach for Evaluating the Performance of Different Designs for a Pilot-Scale Spray Dryer"

_processes, doi:10.3390/pr11010040_

Round 1

Reviewer 1 Report

In work entitled, “Using particle residence time distributions as an experimental 2 approach for evaluating the performance of different designs 3 for a pilot-scale spray dryer”, the performances of four different designs of a pilot-scale spray dryer was evaluated in the light of particle residence time distributions, solid recovery rate and physical properties of the spray-dried products. The experiments were correctly done and the methods of analysis were consistent in showing that the spray dryers performances could be accessed through measurements of particle residence time distributions. Results, perhaps predictablyshowed that dryer geometry plays an important role in both the particle residence time distribution and the wall deposit/ re-entrainment process. In my understanding the work should be published as it is.

Author Response

We gratefully thank you for your time spent on giving us constructive feedback and helpful suggestions.

Reviewer 2 Report

This study deals with the comparison of four spray driers with different designs. RTD measurements were carried out with red dye, then Tank-In-Series type models characterized the particles’ residence time in the machines. It is a valuable work, however, there are some concerns that should be clarified before publication.

Ln 184 This method is used to call Loss-On-Drying.

Ln 188-189 Which marker belongs to which spray dryer design in Figs. 5 and 7?

Ln 261-266 It should be reconsidered whether the gas flow rate has a truly significant effect on 3 of 4 designs as the standard deviations of the estimated number of tank reactors are pretty high. Please evaluate it statistically.

Ln 280-288 Those fluctuations in RTD measuments are unique. This phenomenon should be understood deeply. What are the negative values at the initial part of the measurements? What happens if other dye quantities are injected into the chambers? Could there be a systematic error in the diffraction measurement method?

Typos:

Ln 290 sata data

Author Response

We gratefully thank you for your time spent on giving us constructive feedback and helpful suggestions. Each suggested revision and comment brought forward by you has been incorporated and considered. Below your comments, we have responded to them point by point in detailed responses attached. A different referencing style (i.e., author, date instead of numbering) to the manuscript has been used in the document for clarity.

Reviewer 3 Report

In my opinion the article is very interesting and very well written. My question is how was the zeta potential studied? What medium were the particles suspended in?

Author Response

We gratefully thank you for your time spent on giving constructive feedback and helpful suggestions. Each suggested revision and comment brought forward by you has been incorporated and considered. Below your comments, we have responded to them point by point.

Reviewer #3:

In my opinion the article is very interesting and very well written.

Comment 1:

My question is how was the zeta potential studied?

Reply:

Thank you for your comment. During the spray drying process, the particles are dried in contact with the drying medium (in this case, air), and we believe zeta potential is not applicable in the case of spray drying. Therefore, we did not study the zeta potentials in this study.

Comment 2:

What medium were the particles suspended in?

Reply:

Thank you for your comment. The drying medium we used in this study is air.

Reviewer 4 Report

The submitted manuscript presents an experimental study concerning particle RTD to evaluate the performance of pilot-scale spray dryers design. The manuscript is quite clear and can be accepted after a couple of formal aspects are adjusted as follows:

- Please check coherence of units along the manuscript (e.g. paragraph 2.2 L/min and mL/min)

- Please check some typing aspects 

- Three references that could be interesting to include are suggested:
*Oakley, D. E., 2004. Spray Dryer Modeling in Theory and Practice. Drying Technology 22(6), 1371-1402.
*Di Pretoro, A.; Manenti, F., 2020. Spray Drying. In “Non-conventional Unit Operations: Solving Practical Issues”, Springer, pp. 65-74.
*Anandharamakrishnan, C.; Gimbun, J., Stapley, A. G. F., Rielly, C. D., 2010. A Study of Particle Histories during Spray Drying Using Computational Fluid Dynamic Simulations, Drying Technology 28(5), 566-576.

Author Response

We gratefully thank you for your time spent on giving constructive feedback and helpful suggestions. Each suggested revision and comment brought forward by you has been incorporated and considered. Below your comments, we have responded to them point by point in detailed responses attached. A different referencing style (i.e., author, date instead of numbering) to the manuscript has been used in the document for clarity.

Round 2

Reviewer 2 Report

The authors responded to the comments very carefully. The manuscript can be accepted as it is.